Machine-learning-based quantitative estimation of soil organic carbon content by VIS/NIR spectroscopy

Ding Jianli 1 2
Yang Aixia 1 3
Wang Jingzhe 1 2
Sagan Vasit 4
Yu Danlin yud@mail.montclair.edu 5 6
1 Key Laboratory of Smart City and Environment Modelling of Higher Education Institute, College of Resources and Environment Sciences, Xinjiang University , Urumqi , China
2 Key Laboratory of Oasis Ecology, Xinjiang University , Urumqi , China
3 College of Resources and Environment Science, Qinzhou University , Qinzhou , China
4 Department of Earth and Atmospheric Sciences, Saint Louis University , St. Louis , MO , United States of America
5 Department of Earth and Environmental Studies, Montclair State University , Montclair , NJ , United States of America
6 School of Sociology and Population Studies, Renmin University of China , Beijing , China
Becker Richard
Electronic publication date: 2018 Oct 17
Publication date: 2018
Volume: 6
Electronic Location ID: e5714
Received 2018 May 15; Accepted 2018 Sep 10
Copyright: ©2018 Ding et al.
Copyright year: 2018
Copyright holder: Ding et al.
License: This is an open access article distributed under the terms of the Creative Commons Attribution License, which permits unrestricted use, distribution, reproduction and adaptation in any medium and for any purpose provided that it is properly attributed. For attribution, the original author(s), title, publication source (PeerJ) and either DOI or URL of the article must be cited.
License URL: https://creativecommons.org/licenses/by/4.0/

Keywords: Ebinur lake wetland, Desert wetland soil, Soil organic carbon, Machine learning

Funding: National Natural Science Foundation of China 41771470 U1603241 This work was supported by the National Natural Science Foundation of China (No. 41771470 and No. U1603241). The funders had no role in study design, data collection and analysis, decision to publish, or preparation of the manuscript.

==============================
Soil organic carbon (SOC) is an important soil property that has profound impact on soil quality and plant growth. With 140 soil samples collected from Ebinur Lake Wetland National Nature Reserve, Xinjiang Uyghur Autonomous Region of China, this research evaluated the feasibility of visible/near infrared (VIS/NIR) spectroscopy data (350–2,500 nm) and simulated EO-1 Hyperion data to estimate SOC in arid wetland regions. Three machine learning algorithms including Ant Colony Optimization-interval Partial Least Squares (ACO-iPLS), Recursive Feature Elimination-Support Vector Machine (RF-SVM), and Random Forest (RF) were employed to select spectral features and further estimate SOC. Results indicated that the feature wavelengths pertaining to SOC were mainly within the ranges of 745–910 nm and 1,911–2,254 nm. The combination of RF-SVM and first derivative pre-processing produced the highest estimation accuracy with the optimal values of Rt (correlation coefficient of testing set), RMSEt and RPD of 0.91, 0.27% and 2.41, respectively. The simulated EO-1 Hyperion data combined with Support Vector Machine (SVM) based recursive feature elimination algorithm produced the most accurate estimate of SOC content. For the testing set, Rt was 0.79, RMSEt was 0.19%, and RPD was 1.61. This practice provides an efficient, low-cost approach with potentially high accuracy to estimate SOC contents and hence supports better management and protection strategies for desert wetland ecosystems.

Introduction

Wetlands account for a significant portion of global carbon stocks (Hu et al., 2010). According to the United Nations Environment Programme’s (UNEP) World Conservation Monitoring Centre, the total area of wetlands is about 6% of the total land area globally. Carbon stocks within the wetlands accounts for 14% of the entire land ecosystems (Foley et al., 2005). Due to its high carbon storage, any slight change in wetland carbon stocks might result in significant effect on global climate change (Wang, Zhang & Haimiti, 2015). For example, changes in wetland carbon stocks can increase carbon dioxide concentration and methane in the atmosphere, which might lead to more severe global warming (Pott & Pott, 2004).

Wetlands in arid and semi-arid regions play an important role as ecological barrier in desert ecosystems. Unlike wetlands in wet regions, arid wetlands are highly sensitive to human activities, and the restoration and rehabilitation of them often are extremely hard once degraded (Zhao et al., 2009). Therefore, inland wetlands in arid regions are small but important component that cannot be ignored, especially in global carbon cycle and balance of atmospheric greenhouse gases (GHG) studies (Cole et al., 2007; Liu et al., 2010). Over the past decades, environmental variables (e.g., mean annual precipitation and temperature), soil characteristics including texture and lithology, and the increasingly intensive human activities, like water diversion, reclamation, overgrazing, pollutant emissions have profoundly changed the arid region’s wetland distribution and therefore the balance of carbon budget (Ding & Yu, 2014; Thakur et al., 2012). It is imperative to develop efficient, fast, and relatively accurate approaches to detect, monitor and predict soil organic carbon (SOC) content over large areas in arid regions (Jaber & Al-Qinna, 2011; Stevens et al., 2010; Vohland et al., 2011).

Traditional approaches measuring SOC content employ typical soil chemical analysis methods, mainly including dry combustion techniques (Craft, Seneca & Broome, 1991), chemical oxidation method (West & Post, 2002) and acid solution extraction (Polglase, Jokela & Comerford, 1992). Though those traditional approaches are relatively accurate and widely accepted, they require extensive lab work and often destroy the samples during processing, which renders repeating the lab work nearly impossible. On the other hand, recent studies have reported that the Visible/Near Infrared (VIS/NIR) spectroscopy is a rapid, cost-effective, quantitative and non-destructive technique which provides spectral information with large amount of data to monitor and detect soil quality and chemical components (Kinoshita et al., 2012). Due to the large amount of spectral information, scholars have attempted to establish a wide range of empirical models to seek in-depth understanding between various soil chemicals (such as organic phosphorous, organic carbon, among many others) and the reflective spectra obtained from the spectroscopy analysis (Viscarra Rossel et al., 2006). Among them, machine learning algorithms with their capability to relatively quickly and accurately analyze large amount of data, stand out to provide excellent opportunities for taking advantage of the spectral information (Kuang, Tekin & Mouazen, 2015; Nawar & Mouazen, 2017). The analysis can be done both within the laboratory and in the field.

SOC was one of most important controlling components of soil spectral features. With SOC content of 2% as a boundary, that is, when SOM content exceeded 2%, the SOC played a principal role in masking out the spectral features, while the SOC content was less than 2%, it became less effective (Wang et al., 2017). Many studies have used VIS/NIR spectroscopy to study, estimate and monitor SOC content, but mainly on Alfisols, Entisols, Ultisols (Chang et al., 2001; Summers et al., 2011; Vasques, Grunwald & Harris, 2010), and Mollisols (Araújo et al., 2015; Hong et al., 2018). In general, the conventional regression methods were sufficient for the spectral detection of soil types with higher SOC content. The composition, structure and sedimentary environment of wetland soils were extreme complex, especially in arid regions (Kayranli et al., 2010). Therefore, very few were conducted on wetland soils in arid regions. In addition, most of the empirical approaches focus primarily on applying multivariate linear regression, partial least square regression, or regression kriging to establish relationships between spectra and SOC (Dai et al., 2014; Guo et al., 2015; Liu, Zhang & Zhang, 2008; St. Luce et al., 2014). These approaches often suffer from autocorrelation, nonlinearity, or in some cases overestimation (Wang et al., 2017). On the other hand, machine learning approaches in recent years have gained momentum due to their relative flexibility in adapting (learning) the data structure prior to making any sensible prediction or simulation (McDowell et al., 2012; Peng et al., 2014; Viscarra Rossel & Behrens, 2010; Were et al., 2015).

Applications of various machine learning approaches, such as Support Vector Machine (SVM) and random forest (RF), have been attempted (Meng & Dennison, 2015; Nauman, Thompson & Rasmussen, 2014; Shi et al., 2013). SVM is a powerful calibration method based on the kernel learning methods, it could offer a possibility to train generalizable, nonlinear classifiers in high dimensional spaces using a small training set is that it offers a possibility to train generalizable, nonlinear classifiers in high dimensional spaces using a small training set (Mountrakis, Im & Ogole, 2011; Vapnik, 1999). Differing from existing linear and non-linear regression modeling methods, RF has acceptable predicting performance even if most independent variables are noise (Svetnik et al., 2003; Wang et al., 2018). For instance, Viscarra Rossel & Behrens (2010) and Were et al. (2015) collected soil samples from Kenya and Australia, and applied NIR spectroscopic analyses on the samples. Their study found that both support vector machine and random forest provide reasonably good estimation for SOC. Peng et al. (2014) also applied VIS/NIR spectroscopy with SVM to estimate SOC contents with samples from the middle and lower reaches of the Yangtze River, China. All these researches reported promising results in combining both machine learning approaches and VIS/NIR spectroscopic analysis. Studies on SOC measurement in arid wetlands predominantly employed traditional chemical analytical approaches (Anne et al., 2014; Cohen, Prenger & DeBusk, 2005; Wang et al., 2016; Wang, Zhang & Haimiti, 2015) and the use of machine learning algorithms with spectroscopy are limited.

At present, more machine learning algorithms have been proposed and used for variable selection, e.g., interval partial least squares (iPLS), ant colony optimization (ACO). The general success of combining VIS/NIR spectroscopy analysis and machine learning approaches in other regions calls for in-depth investigation on applying similar approaches in wetlands of arid regions. The combination of two algorithms could maximize the superiority of single method and overcome some faults, to a certain extent. In terms of the Ant Colony Optimization-interval Partial Least Squares regression (ACO-iPLS), it could exhibit certain advantage in distributed parallel calculation, information positive feedback and heuristic search ability (Huang et al., 2014; Zhu et al., 2018). The Support Vector Machine Recursive Feature Elimination (RF-SVM), an intelligent optimization method has demonstrated its outstanding performance and great potential for development in solving many complex optimization problems (Lin et al., 2011). EO-1 Hyperion data with 242 spectral bands was the first satellite borne hyperspectral imaging spectrometer, which has been widely used in soil science, agricultural science, geological mapping, and accurate mapping (Liu et al., 2009). Due to the limitation of atmospheric influence and sensor observation conditions, it was difficult to obtain enough hyperspectral remote sensing imageries to meet the needs of different research fields. Nevertheless, the simulation of Hyperion data based on its spectral response function could serve as the effective substitute for research, when there were no available data. Hence, it is worthwhile to consider these coupling algorithms (RF-SVM and ACO-iPLS) and simulated remote sensed imageries as tools to develop estimating models in soil science.

Given these backgrounds and motivated by previous research, this study aims to combine VIS/NIR spectroscopy and machine learning approaches to quantitatively estimate SOC content of wetlands in arid regions. The study area, Ebinur Lake Wetland National Nature Reserve, located in the arid Northwestern China. The arid wetland ecosystem is very fragile due to the specific climatic conditions. Regional SOC is more sensitive to the climatic changes and human activities than in other areas. In addition, since the study area is located along the Silk Road Economic Belt, its ecological stability has profound significance on the sustainability of local economies as well as the entire Economic Belt (Tan et al., 2018; Wang et al., 2018; Xu et al., 2017). We collected 140 soil samples at various depth in 2012 and analyzed the samples with chemical analysis. In the meantime, we also obtained the spectra information through VIS/NIR spectroscopy. We choose three efficient machine learning approaches, namely, the ACO-iPLS, RF-SVM, RF, to extract feature wavelengths from the spectral data and simulated EO-1 Hyperion data, and further construct the adequately stable and reliable models for the SOC content in the arid wetland regions.

Material and Methods

Study area

Ebinur Lake wetland is a typical arid region lake wetland in Xinjiang Uyghur Autonomous Region, China (44°30′∼45°09′N, 82°36′∼83°50′E). The study area is a combination of lake, river and swamp wetlands, which is ideal for studying SOC contents in arid region (Li, Zhao & Li, 2018). The wetland is located in the northern slopes of the Tianshan Mountains, southwest of the Junggar Basin. The area is surrounded by mountains in the south, west and north, but connect to the Mutetaer desert in the east (Fig. 1). It is a designated eco-protection region, with a land area of 2670.85 km2. The climate is typical temperate arid continental climate with limited annual precipitation (90.9 mm), but very high evaporation (3,400 mm). The annual average temperature is 8.3 °C (Abuduwailil, Zhaoyong & Fengqing, 2015). According to the World Reference Base for Soil Resources (WRB), local prevalent soil types are mainly Arenosols, Solonetz, and Solonchaks (He et al., 2015; Wang et al., 2018). The existence of various typical arid region soil types provides a good opportunity to test the proposed machine learning algorithms’ effectiveness to monitor and evaluate SOC content.

Figure 1 Study area and locations of sampling points.

Vectorization by Jingzhe Wang.

Soil collection and chemical analysis

The soil samples were collected from a field trip to the Ebinur Lake Wetland National Nature Reserve in October 2012. The sampling sites were previously established for various soil properties monitoring purposes. They are located around relatively accessible locations in Kekebasitao Management Station, Yaziwan Management Station, Beidi Management Station, Bird Island, Bortala River Lake estuary, and the lower reaches of Kuitun River. There were in total 35 sampling sites (Fig. 1). At each sampling site, samples were collected at four vertical depths (5 cm, 20 cm, 40 cm, and 60 cm) and five evenly distributed points with a grid of 30 × 30 m (because the spatial resolution of Hyperion imagery is 30 m). The samples for the five points (at each depth) were then mixed evenly to represent the soil for that sampling site (at the specific depth). A total of (4 × 35) samples were collected and then brought to the laboratory for chemical measurements. All soil samples (n = 140) were sufficiently air-dried, ground and sieved through a two mm mesh to remove plant materials, residues, roots and stones. The potassium dichromate method was employed for the measurement of SOC content.

Spectral measurements and pre-processing

The reflectance spectra of all soil samples were measured in the laboratory via an ASD FieldSpec®3 portable spectroradiometer (Analytical Spectral Devices, Inc., St, Boulder, CO, USA) with a wavelength range of 350–2,500 nm. The spectral readings were interpolated to a 1 nm interval. Using recommendation by Zhou et al. (2005), the spectra of all soil samples were measured in a dark room with a 50-W halogen lamp as the light source, which was positioned 0.5 m away from the soil sample, with a 25° zenith angle. The soil samples were put in a 12 cm diameter with 1.8 cm depth container evenly. The optical probe was installed about 0.1 m above the soil sample (Shi et al., 2014). Prior to the first scan, a standardized white Spectralon® panel with 99% reflectance was used to convert radiance to reflectance. To eliminate random reflectance errors, 10 spectral measurements for each sample were taken and the average of these measurements was used as the final spectral reflectance.

High frequency random noises, baseline drifts, and scattering noises could affect spectral measurements. To remove the influence of these noises, Savitaky-Golay (SG) smoothing was implemented with a window size of 5 and polynomial order of 2 via Origin Pro software version 9.0. In general, the transformation of first order derivative was used for the enhancement of the spectral characteristics (Savitzky & Golay, 1964). In this study, the SG preprocessed spectral data was transformed into first order derivative (A′), the inversion of the first order derivative (1∕A′), and logarithm transformation of the first order derivative (lg(A′)). In spectral analysis, they are effective pretreatments, which could eliminate the background noise to a certain extent, and enhance the spectral characteristics (Wang et al., 2017).

Model calibration, evaluation, and comparison

Considering the Euclidean distance of each sample, all 140 soil samples were separated into two equal parts (training set and testing set) using the Kennard-Stone (K-S) algorithm. Each set consists of 70 samples. To investigate the feasibility of using VIS/NIR to predict SOC content and select the most effective pre-processing methods, three machine learning approaches, i.e., Ant Colony Optimization-interval Partial Least Square (ACO-iPLS), Recursive Feature Elimination based on Support Vector Machine (RF-SVM), and Random Forest (RF), were applied for the reduction of inefficacious information and model construction.

ACO-iPLS

ACO-iPLS approach is a combination between principal component analysis based PLS and the meta heuristic optimization Ant Colony (ACO). PLS has been proven a robust and reliable approach in spectral quantitative research, primarily because of its advantages regarding dimension reduction and the synthesis and solving of multi-collinearity problems among independent variables. ACO, on the other hand, is an optimization algorithm that originates from the observation of the ants’ food-seeking behavior in which each ant will leave certain amount of pheromone on the route to the food. A colony of ants would leave enough amount of pheromone to guide the colony to follow the optimal route (with the largest amount of pheromone) to the food. A combination of ACO and PLS algorithms seems to produce fairly useful information to select the most informative spectra or segments of spectra (Huang et al., 2014). Detailed steps are as follows:

(1) Initialization: set the size of the colony (k), for the m segments of spectra, the initial pheromones τi are all set to be 1. (1) τi=1,i=1,2,…,m.

(2) Determine the probability that one segment will be selected in the “route”. The selection is done with the Roulette wheel method, namely, suppose that the time t, segment i’s pheromone is τi(t), then the probability that segment i will be selected is: (2) Pit=τit∑i=1mτit,i=1,2,…,m.

(3) Target function: the prediction accuracy of a PLS model will be used as the target function, specifically, the inversion of the root mean squared error (RMSE) is used here: (3) F=Q∕1+RMSE=Q1+1n ∑i=1nyi−yi∧.

where n is the number of samples, yi is the actual value (SOC content in this study), and yi∧ is the predicted value, Q is a constant that represents the significant factor (significance level). Smaller RMSE indicates a better model.

(4) Update: assume ρ ∈ (0, 1) is the information decaying rate (which can be selected based on empirical studies), then the pheromone of segment i will update as: (4) τit+1=1−ρ×τit+ρ×F.

Following these simple steps, we will iterate steps 2–4. Once the iteration reaches certain amount, the algorithm will produce an optimal set of segments that contains the largest amount of pheromone (information), which also produces the least overall RMSE values.

The SOC contents were regarded as yi. The soil spectra were divided in segments as the different number of routes. The initial number of ants was set to 50, maximum recursive attempts for the PLS to be 50, and maximum iteration to be 20. Based on the minimum RMSE, ρ was set to 0.53, and Q to be 0.01. The threshold of segment was set to 0.3. For the PLS model, the initial number of segments of spectra was set to 15.

RF-SVM

SVM is a machine learning algorithm that has recently attracted quite some attention in dealing with large amount of data. The basic principle of SVM is to use a kernel function that can maximally separate the different classes (González Costa et al., 2017; Thissen et al., 2004). By using non-linear kernels, SVM can essentially map the seemingly inseparable data points to higher dimension, to find the inherent structure of the data. Detailed steps follow:

(1) Set the training samples X0 = [x1, x2, x3…xi, …, xn]T, and the corresponding class labels: Y = [y1, y2, y3, …yi, …, yn]T. Each xi (i = 1, …, n) is a vector containing the spectral information obtained from the VIS/NIR spectroscopic analysis. Each yi is a measured SOC from chemical analysis.

(2) Initialize the feature subset vector s = [1, 2, 3, …, k]. The initial k is the total number of wavelengths (k = 2,151, from 350 nm to 2,500 nm).

(3) Start the iteration: obtain a new training sample based on the remaining features: X = X0 (:, s). The initial training sample includes all 2,151 wavelengths (features).

(4) Use the training sample in SVM and obtain the weight vector (w) for all the features.

(5) Set the ordering rule: c = w2 (to consider only the magnitude of the weights instead of their signs).

(6) Eliminate the feature that has the lowest ordering score C, and then update the training sample.

(7) Repeat steps (3)–(6) until the set s is an empty set to obtain the final result with each feature (wavelength) being ordered.

During each iteration of the RF-SVM algorithm, the lowest ordered feature will be eliminated first, and the remaining features then be re-trained to obtain a new weight vector for next round of feature ordering. For individual features, RF-SVM might not produce optimal results. For groups of features, however, RF-SVM has the potential to produce the best supplementary combinations of features. In practice, the SOC contents obtained from chemical analysis were used as the class labels Y. The spectral information was used as the feature input for the support vector machine X0. The algorithm was implemented with the libsvm-3.1 [FarutoUltimate3.1 Mcode] package based on MATLAB® software version R2015a (MathWorks, Inc., Natick, MA, USA). The optimization was done with a two-dimensional grid searching. With multiple trainings and experiments, we finally decided to use the epsilon-SVM model. The kernel function was selected as Sigmoid. The Gamma value and Eps were set to 0.0039 and 0.01, respectively, and the tuning parameter C was 1.

Random forest

Random forest is an ensemble learning technique developed by Breiman (2001) to improve the regression trees method (Mutanga, Adam & Cho, 2012). In RF regression, the procedure uses bootstrapping to repetitively generate subsamples from a training dataset and train a tree from each subsample. Averaging through these trees results in large gain in reduced variance. As in the previous section, predictors x represents the spectral data, while y represents the actual SOC content in this study, the Abhisheck Jaiantila’s randomforest-matlab package was applied for the implement of random forest learning (Liaw & Wiener, 2002). In the present study, m, the number of sub-samples of the predictors was set to 500.

Model evaluation and comparison

All the above three algorithms were validated via 10 folds cross-validation. The cross-validation correlation coefficient (Rcv), and root mean square error of the cross validation (RMSEcv) were used to optimize all the model parameters. Precision indices of Rcv, RMSEcv, and residual prediction deviation (RPD) were also used to evaluate the performance of these algorithms. Higher Rcv and lower RMSEcv indicate a more stable model. Similarly, for the testing set, higher the Rt, lower the RMSEt suggest better performance. The RPD was used to assess the accuracy of the algorithm. If RPD ≥ 2, then the algorithm has very reliable prediction accuracy. If 1.4 ≤ RPD < 2, then the prediction accuracy is acceptable. Only if RPD < 1.4, the prediction accuracy is not acceptable (Chang et al., 2001). In addition, the deviation of the scatterplot with predicted and measured SOC content from the 1:1 diagonal line can also be used to evaluate a particular algorithm’s prediction accuracy. Intuitively, higher accuracy prediction algorithm will have a better fit along the 1:1 line (Luan et al., 2013).

Hyperion simulation

Despite advances in algorithm development, successful applications of satellite-based methods are limited due to the relative unavailability of sensors with both fine spectral and spatial resolution. The next generation multispectral and hyperspectral sensors such as NASA’s Hyperspectral Infrared Imager (HyspIRI) attempt to address these issues with both increased spatial and spectral resolution but are not yet available. No Hyperion hyperspectral data were available for the study area during the study period. Considering Hyperion shares similar spectral and spatial characteristics with HyspIRI, we simulated Hyperion soil reflectance spectra by using the spectral response function shown in Eq. (5). (5) ρHyperionλ=∫λ minλ maxfλρλdλ∫λ minλ maxfλdλ

where ρHyperion is the simulated Hyperion reflectance spectra of band λ; f(λ) is the spectral response function of the simulated band λ; ρ(λ) is the measured reflectance spectral at band λ; and λmin and λmax are the lower and upper bounds of measured reflectance spectra, respectively (Maimaitiyiming, Miller & Ghulam, 2016).

Results

Descriptive statistical analysis of SOC content

The statistical characteristics of both the training and testing sets were shown in Table 1. The standard deviations between the training and testing sets were similar, and differences between the average value was reasonably small. The fact suggests that the selection of both data sets was representative. The spectral curves of soil samples with different SOC contents were illustrated in Fig. 2, overall spectral reflectance increases as the SOC contents decrease. The diagram shows that SOC contents of less than 1% and more than 2.5% correspond to the highest and lowest reflectance, respectively. The pattern of the spectral reflectance fluctuation remains similar regardless of the different SOC contents. Specifically, the reflectance tends to be low in the visible bands (350–780 nm), but high in the infrared bands (780–2,500 nm). The significant absorption features were in the range of 1,850–1,950 nm, and the spectral curves increased rapidly to the peak of 2,100 nm.

Table 1 Descriptive statistics of soil organic carbon in both training and testing sets.

Models	Sample size	Min/%	Max/%	Mean/%	St.dev/%	
Training sets	70	0.02	2.97	0.51	0.64	
Testing sets	70	0.01	3.42	0.40	0.65	

Figure 2 Spectral reflectance of different wetland soil organic carbon contents.

(A) Arenosols, (B) Solonetz, (C) Solonetz, (D) Solonchaks, (E) Solonetz.

Comparing the results from this study with results reported in other regions (Hong et al., 2018; Peng et al., 2014; Viscarra Rossel et al., 2006), we observe that the spectral reflectance of soils in arid wetland regions were similar to those of agricultural soils, though in the range from 1,900 to 2,100 nm, spectral reflectance of arid wetland soil increases faster than that of the crop land.

Results of ACO-iPLS algorithm

Although spectral data contains very large amount of data, not all of them are necessarily informative towards detecting SOC contents (Stenberg et al., 2010). If all wavelengths were applied in the construction of the models, some irrelevant spectral data could be included, and yields inferior estimation accuracies (Liu, Zhang & Zhang, 2008). Thus, the model based on informative spectra could outperform those using all wavelengths.

For that reason, the ACO-iPLS algorithm first selected the informative spectra segments (after pretreatments), and then relevant spectra segments were entered to the model to predict the SOC content. The selected wavelengths and the modeling results with ACO-iPLS were reported in Table 2. It was obvious that the original first order derivative transformation with selected wavelengths yielded the best model performance with the optimal Rcv and RMSEcv of 0.87 and 0.33%, respectively. For the testing set, Rt was 0.83, RMSEt was 0.40%, and the RPD was 1.63, indicating reasonable prediction. Other transformations yield rather low RPD scores (0.87 and 1.10, respectively).

Table 2 Selected feature wavelengths, training sets and testing sets results by ACO-iPLS method.

Pre-processing	Selected wavelengths	Training sets	Testing sets	
		Rcv	RMSEcv	Rt	RMSEt	RPD	
A′	1,786∼1,929	0.86	0.33	0.83	0.40	1.63	
1∕A′	494∼638	0.64	0.57	0.76	0.74	0.87	
lgA′	1,786∼1,929	0.73	0.50	0.82	0.59	1.10	

Results of RF-SVM

We applied the RF-SVM algorithm to quantify the SOC contents using selected wavelengths via the three transformations. The selected wavelengths and modeling results were reported in Table 3. The result again suggested that the original first order derivative transformation yielded the best performance. Rcv is 0.97, RMSEcv is 0.16%. For the testing set, the Rt is 0.91, RMSEt is 0.27%, and RPD is 2.41. The results indicated the RF-SVM algorithm recorded overall better model performance. Even for the other two transformations, the RPDs also reached 1.88 and 1.44, indicating that the algorithm could give reasonable prediction.

Table 3 Selected feature wavelengths, training sets and testing sets results by RF-SVM method.

Pre-processing	Selected wavelengths	Training sets	Testing sets	
		Rcv	RMSEcv	Rt	RMSEt	RPD	
A′	780, 1,911, 783, 779, 768, 759, 793, 794, 2,254, 910, 1,677, 1,912, 2,089, 745, 825, 2,088, 746, 2,090, 1,913, 1,751	0.97	0.16	0.91	0.27	2.41	
1∕A′	663, 1,836, 658, 2,431, 2,494, 618, 999, 746, 370, 2,475, 960, 510, 1,081, 443, 1,681, 1,123, 360, 793, 2,123, 2,476	0.99	0.03	0.84	0.34	1.88	
lgA′	706, 736, 731, 1,943, 779, 721, 413, 510, 704, 397, 732, 1,944, 1,085, 2,091, 2,347, 881, 2,422, 1,966, 2,257, 2,111	0.99	0.03	0.81	0.45	1.44	

Results of RF

Based on the three transformations of spectral data, the selected feature bands and RF modeling results were reported in Table 4. Again, the original first order derivative transformation yields the best performance, Rcv was 0.98 and RMSEcv was 0.15%. For the testing set, the Rt is 0.92, RMSEt is 0.33%, and RPD is 1.98. The algorithm also yielded very good prediction. Even for the other two transformations, the RPDs are 1.51 and 1.58, indicating reasonably acceptable prediction accuracy.

Table 4 Selected feature wavelengths, training sets and testing sets results by RF method.

Pre-processing	Selected wavelengths	Training sets	Testing sets	
		Rcv	RMSEcv	Rt	RMSEt	RPD	
A′	794, 740, 758, 713, 741, 821, 789, 766, 613, 682, 732, 776, 822, 720, 769, 746, 635, 733, 940, 668	0.98	0.15	0.92	0.33	1.98	
1∕A′	1,403, 1,402, 1,390, 1,399, 1,405, 1,404, 2,189, 2,196, 620, 2,176, 822, 2,192, 809, 2,177, 670, 632, 2,191, 1,388, 727, 2,315	0.98	0.14	0.83	0.43	1.51	
lgA′	676, 633, 2,189, 2,202, 2,195, 675, 1,402, 2,183, 722, 632, 620, 703, 821, 2,205, 2,193, 689, 2,200, 646, 812, 714	0.98	0.14	0.90	0.41	1.58	

From Tables 2–4, it was evident that the original first order derivative transformation of the spectral information yielded the most reasonable modeling performance among all three algorithms. Our discussion will then focus on modeling results based on this transformation only. Figures 3–5 are the results of selected feature wavelengths under the first order derivative transformation. Table 5 lists the modeling performance using wavelengths after first order derivative transformation with both the training and testing sets.

Figure 3 Selected spectral interval by ACO-iPLS with first derivative spectra.

Figure 4 Selected wavelengths by RF-SVM with first derivative spectra.

The ACO-iPLS algorithm subdivided the entire wavelength (350–2,500 nm) into 15 segments (Fig. 3). The 11th segments (1,786–1,929 nm) yielded the best performance with the lowest RMSE value. From Fig. 4, based on RF-SVM algorithm, the optimal wavelengths were in the segments of 745–910 nm and 1,911–2,254 nm. Based on mean decrease in accuracy (Fig. 5), RF selected wavelengths ranging from 613 to 940 nm. Combing all three selection results, the optimal wavelengths that were most relevant to SOC content are located within segments of 745–910 nm and 1,911–2,254 nm for the arid Ebinur Lake wetland soils.

Figure 5 Selected wavelengths by RF with first derivative spectra.

Table 5 Comparison of the results by different models with first derivative spectra.

Modeling methods	Training sets	Testing sets	
	Rcv	RMSEcv	Rt	RMSEt	RPD	
AOC-iPLS	0.86	0.33	0.83	0.40	1.63	
RF	0.98	0.15	0.92	0.33	1.98	
RF-SVM	0.97	0.16	0.91	0.27	2.41	

From Table 5, we learn that the RF-SVM produces the highest RPD (≥2) among the three algorithms, followed by RF and ACO-iPLS. In addition, Fig. 6 shows scatterplots of predicted and measured SOC contents. The slope for the RF-SVM model were well distributed on the 1:1 line indicating the best fit which further confirms the above observations.

Simulation application to satellite data

The results of all three algorithms performed on laboratory-derived spectra data showed that RF-SVM approach with the first derivative pre-processing produced the highest estimation accuracy. In this practice, we evaluated the feasibility of simulated Hyperion reflectance spectra to estimate SOC by using RF-SVM approach. The selected wavelengths and the modeling results were illustrated in Table 6. Using the RF-SVM algorithm, the optimal wavelengths were in the segments of 702–824 nm and 2,083–2,426 nm. And there was some difference between the feature bands of Hyperion and those of laboratory-derived data. It was observed that the original first order derivative transformation with selected wavelengths yielded higher Rcv (0.96) and RMSEcv of 0.23%. For the testing set, Rt was 0.79, and RMSEt was 0.19%, the RPD was 1.61 (Fig. 7). The results suggest that VIS/NIR bands of hyperspectral satellite data have a good potential for predicting wetland SOC content in arid areas.

Figure 6 Measured content and the values estimated by SVM model with simulated EO-1 Hyperion data.

Table 6 Selected feature wavelengths and training sets and testing sets results by RF-SVM method with simulated EO-1 Hyperion data.

Modeling methods	Selected wavelengths	Training sets	Testing sets	
		Rcv	RMSEcv	Rt	RMSEt	RPD	
RF-SVM	824, 813, 2,194, 2,426, 2,093, 702, 2,083, 712, 2,436, 803, 2,174, 2,214, 2,163, 2,416, 2,103, 722, 2,133, 1,810, 1,669, 2,123	0.96	0.23	0.79	0.19	1.61	

Figure 7 Comparison of the measured content and the values estimated by different models.

Discussion

Applying spectral techniques to evaluate, monitor and predict SOC content is an important approach, especially in arid regions where SOC is critical to soil quality yet soil sampling often is expensive and sometimes very hard for laboratory tests. Establishing an effective model to take advantage of the large amount of spectra information from VIS/NIR spectroscopy technology is of great interest in the study of SOC content in arid regions. In the current study, we found that the original first order derivative transformation was able to retain the most useful information from spectral data among the three discussed transformations including original first order derivative, inversion of first order derivative, and logarithm of first order derivative. Zornoza et al. (2008) also found similar results in their study, with high coefficient of correlations of 0.95 and 0.98, which was consistent with our results.

The massive information obtained from hyperspectral sensors often requires effective algorithms to produce the best and most robust prediction. The selection of feature spectra is hence a critical step in applying VIS/NIR spectroscopy technique. From Tables 2–4, we could see that the approaches of RF-SVM and RF were able to select 20 wavelength segments, and the ACO-iPLS selected a large chunk of segments that produce the most desired results. Comparing the results of the three algorithms, we observed that the relevant wavelengths concentrated in the range of 745–910 nm and 1,911–2,254 nm. Results obtained in the current study were not in accord with previous research (400–800 nm, 1,030–1,080 nm and 2,250–2,340 nm), this could be attributed to the difference among different soil types (Wang et al., 2016).

In addition, our experiment with the three algorithms also indicated that different algorithms yield different results. Among the three algorithms, the RF-SVM produced the best results, followed by RF algorithm, while the ACO-iPLS algorithm performs least ideally. We argue that the ACO-iPLS algorithm is not necessarily an inappropriate algorithm as the results are still acceptable. However, ACO-iPLS algorithm, a bionic approach attempting to mimic the ants’ intelligence, could yield effective optimization if the amount of information was set to appropriate size. On the other hand, if the amount of information was massive, the complexity of the information might lead to local optimal to dominate the optimization process. Though by further fine-dividing the segments, we might be able to produce better results, we are still at risk of being stuck at potential local optimal with the ACO-iPLS algorithm. Random forest algorithm, on the other hand, uses multiple trees (the forest) to produce enhanced learning outcomes. Because of its flexibility, it often produces the best results with the training set since it can distinguish between the useful information and inevitable noises existed within the training set. The problem, however, as with many tree-related algorithms, is that it can easily slide to over-fitting and might be impacted by the very skewed dataset which is the case for our dataset. The results from the current study support this argument. The RMSEcv of the random forest algorithm for the training set was 0.15, lowest among all three algorithms. While the RMSEt was higher than that of SVM (0.33 versus 0.27, Fig. 6), indicating possible over-fitting issues for the random forest algorithm. SVM is a structural empirical risk model, the parameters of the decision function are determined by empirical analysis. Since the goal of the algorithm when training the parameters was to minimize risks, it allows for some errors during fitting while assigning certain penalty to such errors (by adjusting the tuning parameter, C). This also agrees logically with the fact that there shall be inevitable noises in the training data. Our analysis suggested that the SVM produced results were better. The RPD of the model reached 2.41 while those of RF and ACO-iPLS were 1.98 and 1.63, respectively. The results were in line with Were et al. (2015) and Viscarra Rossel & Behrens (2010) who used similar algorithms to study SOC contents in Kenya and Australia.

The results of simulated Hyperion spectral analysis showed that the feature wavelengths pertaining to SOC were mainly located around the ranges of 702–824 nm and 2,083–2,426 nm. This finding differs from previously reported work by Morgan et al. (2009) that identified the feature wavelengths between 610 and 650 nm for SOC. It is worth noting that the 702–824 nm wavelengths overlap with the findings by Ji, Viscarra Rossel & Shi (2015) who identified the absorption feature at the wavelengths from 600 to 800 nm for SOC. Although the combination of laboratory-derived reflectance with RF-SVM produced slightly better estimation than the simulated Hyperion reflectance spectra, the advantage of the simulated approach is evident, which is critical for the potential uses of the planned hyperspectral sensors soon to be available.

In fact, the simulated hyperspectral was constructed based on the specified spectral response function and field-derived spectral data (Jin et al., 2017; Liu et al., 2009). However, the obtained continuous spectrum information of every pixel was affected by geographical atmospheric, meteorological, and lighting variations, e.g., cloud cover, precipitation, pixel purity, and temporal-spatial resolution of ground target (Hill, 2013; Zhou et al., 2013). Hence, there was a difference between the actual VIS/NIR reflectance and simulated satellite data, the quantitative estimation models were only theoretically valid. To further improve the applicable capability of the established model, the prediction accuracy, combination of various simulated and actual satellite sensor data will be analyzed in the future study. Various spaceborne hyperspectral data or imaging spectroscopy will be available, e.g., ESA’s PROBA, Chinese GaoFen-4, and upcoming Germany’s DLR’s EnMAP, which is very helpful for achievement of the quantitative analysis on remote sensing (Liu et al., 2018). Therefore, actual satellite data should be used in future studies to evaluate the SOM estimation model. We did not consider the effect of the spatial resolution of remote sensing imagery on the estimation accuracy of SOC content. In terms of the actual satellite data, the different radiometric correction and atmospheric correction approaches could result in the changes of spectra of targets. Additionally, the actual satellite data and the simulated satellite data differ because of the measurement errors, signal noises, and atmospheric environment (Maimaitiyiming, Miller & Ghulam, 2016). The accuracy and detection limits of estimations could be affected by these mentioned factors. In future study, more intelligent algorithms will be applied to overcome the scale differences in both spectral and spatial dimension of actual and simulated data.

This study clearly suggests that VIS/NIR spectroscopy is an effective method to detect wetland SOC content of soils in arid regions. Our work provides a comprehensive evaluation of models and algorithms for their power to identify relevant feature wavelengths to estimate SOC. Such endeavor is of critical and practical importance. Increasing population and intensive human activities have put ever increasing pressure on wetland in arid regions. Changes in the SOC content in fragile ecosystems can be drastic even with slight increase of human activities (Câmara et al., 2016; Smith et al., 2016). Such changes could have significant impact on local climatic and ecological systems, and even contribute to large-scale carbon equilibrium (Prasad et al., 2016). The proposed VIS/NIR spectroscopic approach plus the relatively mature classification and prediction models provide effective means to the local ecological and environmental management authorities.

Conclusion

In this study, the first order derivative transformation provides the best predictive power among proposed wavelength transformation strategies (A′, 1∕A′, and lg(A′)). All three algorithms consistently suggest that the wavelength segments of 745–910 nm and 1,911–2,254 nm are the most effective spectral regions to detect SOC content. Among the three models, SVM based recursive feature elimination algorithm produces the best overall results for both the training and testing datasets with an RPD of 2.41. The other two approaches, namely, ACO-iPLS and RF also produce reasonably well results following SVM. In addition, the simulated EO-1 Hyperion data combined with SVM based recursive feature elimination algorithm produces high accuracy of estimating SOC content with an RPD of 1.61. The RF-SVM algorithm identified the wavelength segments of 702–824 nm and 2,083–2,426 nm as the most effective spectral regions to detect SOC content at the satellite level. Overall, the simulated Hyperion data have a great potential for predicting wetland SOC content in arid regions. The proposed combination of VIS/NIR spectroscopy technique and SVM based recursive feature elimination algorithm provides a fast, economic, and robust approach to monitor, detect, and predict SOC contents in the arid and semi-arid region wetlands.

Supplemental Information

Supplemental Information 1 Reflectance of soil samples (n= 140)

Click here for additional data file.

Supplemental Information 2 Modeling code

Click here for additional data file.

We are especially grateful to the anonymous reviewers and editors for appraising our manuscript and for offering instructive comments.

Additional Information and Declarations

Competing Interests

Author Contributions

Data Availability

Danlin Yu is an Academic Editor for PeerJ.

Jianli Ding conceived and designed the experiments, analyzed the data, contributed reagents/materials/analysis tools, authored or reviewed drafts of the paper, approved the final draft.

Aixia Yang conceived and designed the experiments, performed the experiments, analyzed the data, contributed reagents/materials/analysis tools, approved the final draft.

Jingzhe Wang performed the experiments, analyzed the data, contributed reagents/materials/analysis tools, prepared figures and/or tables, authored or reviewed drafts of the paper, approved the final draft.

Vasit Sagan performed the experiments.

Danlin Yu conceived and designed the experiments, analyzed the data, contributed reagents/materials/analysis tools, prepared figures and/or tables, approved the final draft.

The following information was supplied regarding data availability:

The raw data are provided in the Supplemental Files.

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
