# Peer review of "Machine-learning-based quantitative estimation of soil organic carbon content by VIS/NIR spectroscopy"

_PeerJ, doi:10.7717/peerj.5714_

## Round 0.1 · original submission · Major Revisions

We have received two reviews of your manuscript, and there are several major issues that need to be dealt with before this manuscript can be considered for publication.

Both reviewers have provided constructive feedback which will improve the manuscript. You will need to go through all of the reviewers comments and address each of the issues raised one by one in the revised manuscript.

Reviewer 2 has specific concern in the introduction and methods section (generally, experimental design and methods justification) and the conclusion section which must be addressed. These include justification of the specific machine learning methods used, and addressing the points raised about simulating hyperion. These concerns are listed in the experimental design comments, and reiterated in the comments on specific manuscript lines.

There are overall English/proofeading issues that need to be addressed. Both reviewers note note numerous specific instances, but the corrections are not limited to these. I suggest that once other revisions are made, you have the article edited for English.

Reviewer 1 ·

Basic reporting

1) Clear, unambiguous, professional English language used;
2) Both the introduction and the background are well presented and the references listed are also relevant.
3) Structure conforms to PeerJ standard.
4) Figures are relevant, well labelled & described. It is suggested that the author needs to use different colors and different line types to better describe the sample data, preferably color figures.
5) Raw data supplied.

Experimental design

1) Within Scope of the PeerJ journal.
2) Research question well defined, relevant & meaningful. It is stated how the research fills an identified knowledge gap.
3) Structure conforms to technical & ethical standard..
4) Methods described with sufficient detail & information to replicate.

Validity of the findings

1) Data is robust, statistically sound, & controlled. 2) Conclusions are well stated, linked to original research question & limited to supporting results.
3) Speculation is welcome.

Additional comments

I would like to acknowledge the authors for the tremendous amount of work they have put into developing the manuscript. This article uses feature selection algorithms and multivariate modeling methods to estimate SOC content in arid regions. The overall level of English expression is very good, but some detailed parts need to be improved, especially in the section of Material and methods. I recommend acceptance of the manuscript after revision following the specific comments listed below.


Some specific suggestions:
Line 49: please replace “efficacy” with “feasibility”.
Line 52: please replace “SVM-RFE” with “RF -SVM”. Same as below.
Line 54: please replace “mainly were in” with “were mainly within”.
Line 58: The sentence of “For the testing set, Rt 59 was 0.79, RMSEt was 0.19 %, and RPD was 1.61.” is not very clear. It's best to connect with the last sentence.
Line 85: please add “, and ” before “therefore”.
Line 86: Who does “it” refer to? Please revise.
Line 90: replace “conteny” with “content”.
Line 91: please add “including ” after “mainly”.
Line 110: please replace “partial ordinary least square regression ” after “partial least square regression”.
Line 133: please replace “this context ” with “these backgrounds”.
Line 164: please replace “content” with “property”.
Line 172: please replace “selected” with “collected”.
Line 173: please replace “screen” with “mesh”.
Line 189: please remove “also”.
Line 201: better remove “using”.
Line 204: please replace “if” with “is”.
Line 204-206: “PLS if often employed to address heavy multi-collinearity among the variables, which is very common in spectral datasets due to the small increment of wavelengths.” Please rewrite this sentence, it is difficult to follow.
Line 253: Please add a period to the end of the sentence.
Line 262: Please tilt Y.
Line 263: Please revise the sentence, “The algorithm was implemented in were implemented with the …….”.
Line 276: Please revise the sentence, “…… will be used to split the tree each time a split is to happen”.
Line 278-279: Please revise the sentence, “it is hence very possible the influence of the strong predictor(s) is avoided and different trees can be produced.”.
Line 319: please replace “content” with “contents”.
Line 330-331: Please revise the sentence, “……the performance of model could be under threat.”.
Line336: Please add a space in the sentence of “Rt was0.83”.
Line348: please replace “spectra” with “spectral”.
Line 367: Please add a space in the sentence of “we could learn thatthe”.
Line398: please replace “Compared” with “comparing”.
Line398: please replace “concentrate” with “concentrated”.
Line 401-402: Please revise the sentence, “in the spectral wavelengths Hou et al. (2014).”.
Line 453-455: Please revise the sentence, “All three algorithms consistently identified the wavelength segments of 745-910 nm and 1911-2254 nm the most effective spectral regions to detect SOC content.”.
Figure 2: It is better to distinguish five curves by using different colors and different line types.
Figure 4: Please revise the sentence “SVE-RFE”.

·

Basic reporting

This paper has grammatical errors, and unclear uses of English in places. Thorough review for language and readability needs to take place before it can be seriously considered for publication.

References are sufficient and thorough and field background is provided.

The raw data for the spectra is shared, but the SOC contents were not provided. Those are required for anyone to validate if the methods proposed by the authors produces consistent results with this data set.

Statistical methods should be simplified, too much detail currently and could be made cleared with cleared language.

Tables and Figures contain duplicate information and could be simplified and merged to reduce the number of tables and figures. This would make interpretation of the results easier.

Experimental design

The research question is well defined. However, the authors do not justify sufficient why the methods used were selected for this study. They do establish that limited work has happened with these soils, which does merit examination. However, they are other statistical tools that have been used in recent (last 5 years) literature (which I provide in the general comments section) that are not included. The authors need to justify why they used the methods they did.

Additionally, the rationale for the hyperion simulation is not provided. This analysis is not meaningful for a number of reasons. The first is that direct soil measurements require bare soil and the authors did not establish of bare soil with no vegetation cover occurs in this region. Second, satellite measurements have additional factors such as atmospheric effects, varying levels of solar radiation and angle, calibration effects etc. None of these factors were simulated in their lab analysis, which makes their simulation a poor representation of reality. Additionally, new tools such as EnMAP from DLR are planned, and the rationale for simulating hyperion is not established. Hyperion was a test program, and a important one, but it is not the future of hyperspectral satellite imaging. A study designed for the newer proposed tools would be more meaningful.

Validity of the findings

The analysis is robust and the analysis is sound. Particularly the use of cross validation and an independent 50-50 training and test data set selected by the Kennard-stone algorithm was defensible.

However, the conclusion I not well stated and is written more as an abstract. A clear conclusion needs to be written.

Additional comments

Line 56: indicate that Rt is the testing set results
Line 68: Carbon stocks not stock
Line 69: Can delete “Despite its small coverage, the wetlands store nearly 1/6 of all the carbon stock on Earth. “ already cleared stated with previous sentence
Line 77: This sentence is poorly worded and confusing
Line 84: Delete unreasonable, that indicates a value judgement that should not be stated
Line 89: in arid regions, not in the arid regions
Line 90: SOC content not conteny
Line 94: Confusing grammar
Line 95: Have reported that
Line 96: change eco-friendly to non destructive
Line 111: This is true of the historical literature, but not for publication over the past 5 years. There have been a number of machine learning focused papers i.e. (Rossel and Behrens 2010; Chun et al. 2012; Doetterl et al. 2013), some of which are cited in the next paragraph. Need to state that initial approaches for VIS/NIR spectroscopy included the methods stated
Line 117: these papers and others have used a variety of other machine learning tools: ex. Cubist models, ANN, gradient boosting etc. Some of which have had more success than random forest and SVM. Need to justify where SVM and RF were used specifically.
Line 132: Why is this the case? This is an unsubstantiated assertion
Line 139: This seems like a very bold claim. Can the authors provide a reference to justify this statement?
Line 159: Report soil types according to an international soil system, either the WRB or USDA systems
Line 192-194: Justify why these smoothing methods were selected
Line 197: Good justification for training sample selection, and 50-50 split is sufficient and defensible.
Line 199: Need to justify why these particular models were selected
Line 234: How were p and Q selected? Was it is based on RMSE minimization?
Line 271: Mutange reference is wrong, please site correct reference.
Line 284: Should be a new subsection, and explained earlier in the methods.
Line 292: Gao et al 2011 is not the original reference for this convention. See (Chang et al. 2001)
Line 296: I don’t see the justification for the hyperion simulation. It is not a true simulation as this study does not have all the atmospheric and lighting condition variation that comes with satellite data. Additionally newer proposed satellite programs such as EnMAP from DLR will be the future of satellite hyperspectral imaging.
Line 441: Conclusion is too long. Needs to be simplified to one paragraph with clear conclusions and statements from the research. Currently reads like an abstract.

All tables: Caption needs to completely explain the table, including what the abbreviations mean.

Figure 1. Include source for geographic data
Figure 2. Add more explanation about what types of soil samples were used in the caption
Figure 3. Not needed. This information could be merged into table 2
Figure 4. Not needed. Duplicates info in table 3.
Figure 5. Not needed. Duplicates info in table 4.

---

## Round 0.2 · Minor Revisions

The two reviewers agree that the manuscript is much improved, and with a few modifications should be acceptable to publish. Reviewer 1's comment should be straightforward to address. Reviewer 2 accurately points out that being clear about possible differences techniques applied to other soil types might be expected to have with wetlands soils is important. I would also urge you to consider reviewer 2's comment on including a brief justification of method selections in the introductory material, and with adding a brief discussion about the impact of different measurement errors from the Hyperion instrument and the ASD. I agree with the authors that though there is slight duplication in the tables and the figures discussed by reviewer 2 they are appropriate for the paper, and should be included in the paper.

Reviewer 1 ·

Basic reporting

No comment.

Experimental design

No comment.

Validity of the findings

No comment.

Additional comments

The author has made revisions to the questions I have listed. In general, after the revision, the article’s quality has been greatly improved. There is only one suggestion below that needs to be revised:
Line 215-218: “Due to its advantage of dimension reduction, synthesis, and solving multi-collinearity problems among independent variables, PLS has been adopted as a robust and reliable approach in spectral quantitative research.” Please rewrite this sentence.

·

Basic reporting

The article is clearly written, and the number of unclear sentence is limited.

Sufficient literature has been cited.

The article is professional structured. Currently there is duplication in the information in Figures 3, 4 and 5, and Tables 2 to 6. The figures and tables should be simplified to avoid the duplication of information.

The results and hypotheses are relevant and self-contained.

Experimental design

The research is original primary research. The research question is well defined and meaningful but more work is still needed to explain how it fills a knowledge gap. The following needs to be clearer:

The authors needs to explain more clearly why these soils are different any why techniques applied to Alfisols, Entisols and Ultisols and Molllisols won’t necessarily apply to wetland soils. What is different about how wetland soils will interact with the methodology used in the paper needs to be explicitly stated and explained. Wetland soils are under investigated in reflectance spectroscopy, and in soil science in general, the authors need to make the case of what is different about them more explicitly.

Additionally, the rationale for the ACO-iPLS methodology inclusion needs to be discussed and the rationale for the hyperion simulation need should be in the introduction.

The methods are described in sufficient and clear.

Validity of the findings

The findings are valid and clearly explained, and the data is robust and statistically sound. A 50/50 training/testing split is very robust and defensible, and is more robust than typical, so excellent work by the authors in that regard.

The conclusions are more clearly stated, and the manuscript has been improved. One outstanding question for the discussion is how might the signal:noise compare between the field spec 3 and the simulated hyperion data. How does that influence accuracy and detection limits.

Additional comments

The manuscript has been improved. The main outstanding issue is more improvement is needed in how the introduction sets up the rationale for all aspects of the paper.

Also the authors have not provided sufficient justification for continuing to include Figures 3, 4 and 5. The information is duplicated amongst the information with the Tables. The quality of the paper is reduced by duplicating information in the tables and figures as it makes the paper more complicated to read and review.

---

## Round 0.3 · accepted · Accept

Thank you for completing the requested revisions to this manuscript. The additions and clarifications made in both revisions have strengthened and improved the clarity of the presentation.